# Cardiometabolic Therapies Shape Non-Coding RNA Landscapes in Cardiovascular Fibrosis

**DOI:** 10.3390/metabo15100664

**Published:** 2025-10-11

**Authors:** Erica Floris, Francesco Nutile, Claudia Cozzolino, Virginia Pontecorvi, Antonella Bordin, Elena De Falco, Vittorio Picchio, Isotta Chimenti, Francesca Pagano

**Affiliations:** 1Department of Medical and Surgical Sciences and Biotechnologies, Sapienza University of Rome, 04100 Latina, Italy; erica.floris@uniroma1.it (E.F.); nutile.1912482@studenti.uniroma1.it (F.N.); claudia.cozzolino@uniroma1.it (C.C.); virginia.pontecorvi@uniroma1.it (V.P.); antonella.bordin@uniroma1.it (A.B.); elena.defalco@uniroma1.it (E.D.F.); isotta.chimenti@uniroma1.it (I.C.); 2Maria Cecilia Hospital, GVM Care & Research, 48033 Cotignola, Italy; 3Department of AngioCardioNeurology, IRCCS Neuromed, 86077 Pozzilli, Italy; vittorio.picchio@uniroma1.it; 4Institute of Biochemistry and Cell Biology, National Council of Research (IBBC-CNR), 00015 Roma, Italy

**Keywords:** fibrotic remodeling, metabolic modulators, microRNAs, long non-coding RNAs, epigenetic regulation, cardiometabolic syndromes, therapeutic targets

## Abstract

**Background**: Cardiometabolic syndromes, including diabetes, obesity, and metabolic syndrome, significantly contribute to cardiovascular fibrosis, a major driver of heart failure. Non-coding RNAs (ncRNAs)—notably microRNAs (miRNAs), long non-coding RNAs (lncRNAs), and circular RNAs (circRNAs)—have emerged as critical epigenetic regulators of fibrotic remodeling. Recent studies indicate that widely used metabolic modulators can influence ncRNA expression, potentially impacting on cardiovascular fibrosis. This review synthesizes evidence on the interplay between metabolic therapies and ncRNA regulation, with emphasis on therapeutic and biomarker potential of miRNAs. **Methods**: A literature search was manually curated and conducted on PubMed for studies published mainly in the last decade and evaluating the effects of metformin, sodium-glucose cotransporter-2 (SGLT2) inhibitors, peroxisome proliferator-activated receptor gamma (PPARγ) agonists, glucagon-like peptide 1 (GLP-1) receptor agonists, and fatty acid oxidation inhibitors on ncRNA expression in the context of cardiovascular fibrosis. Data from in vitro, in vivo, and clinical studies were extracted and categorized by drug class, ncRNA target, and functional outcomes. **Results**: Several metabolic modulators specifically downregulate pro-fibrotic (miR-21, miR-92, H19, and metastasis associated lung adenocarcinoma transcript 1 (MALAT1)) and upregulate anti-fibrotic ncRNAs (miR-29, miR-133a, miR-711, miR-133a, miR-30a and miR-200 family). This results in global attenuation of the transforming growth factor- beta (TGF-β) signaling, which limits extracellular matrix (ECM) accumulation thus improving myocardial compliance. Across drug classes, changes in ncRNA profiles paralleled improvements in fibrosis-related endpoints. **Conclusions**: Metabolic modulators exert anti-fibrotic effects partly through ncRNA regulation, offering novel therapeutic strategies and potential biomarkers for cardiovascular fibrosis in cardiometabolic disease. Targeting metabolic–ncRNA crosstalk may enable more precise and synergistic interventions for preventing or reversing pathological remodeling.

## 1. Introduction

Cardiometabolic syndromes contribute significantly to heart failure and cardiovascular fibrosis. Traditional therapies target systemic symptoms like hypertension or hyperglycemia, but these approaches have limited impact on fibrotic remodeling of the heart. The advent of epigenetic and transcriptomic insights has brought non-coding RNAs (ncRNAs) into focus as crucial regulators and promising therapeutic targets [1,2,3]. 

Research on the role of ncRNAs, particularly microRNAs (miRNAs), long non-coding RNAs (lncRNAs), and circular RNAs (circRNAs), in cardiovascular fibrosis is rapidly expanding. These ncRNAs regulate gene expression and are intimately involved in the development and progression of fibrotic remodeling of the cardiac tissue [2]. Cardiometabolic conditions such as diabetes, obesity, and metabolic syndrome intensify cardiovascular fibrosis [4,5,6], often through dysregulation of ncRNAs [7]. Several recent studies highlight the therapeutic potential of modulating ncRNAs, either by targeting their expression or through delivery via extracellular vesicles [8]. Epigenetic therapies and exercise-mimetic interventions involving ncRNAs are also being developed [9,10].

Cardiovascular fibrosis involves the pathological accumulation of extracellular matrix (ECM) components, leading to increased myocardial stiffness and impaired cardiac and vascular function. Cardiovascular fibrosis is a pathological hallmark shared by a wide spectrum of cardiovascular diseases, including heart failure, arrhythmias, hypertrophic cardiomyopathy, diabetes, and others [11]. Despite its strong association with adverse outcomes, no therapy currently exists that specifically targets cardiac fibrosis itself, and available treatments mainly act on the underlying cardiovascular condition or related mechanisms [12]. Furthermore, as fibrosis rarely occurs in isolation and is almost invariably accompanied by other cardiovascular pathologies, it is not possible to accurately estimate survival or mortality rates attributable solely to cardiac fibrosis.

While metabolic disturbances are well described contributors to the cardiovascular fibrosis establishment and progression, emerging evidence underscores the role of ncRNAs in mediating fibrotic responses [13]. Interestingly, it appears that metabolic modulators can influence ncRNA expression, offering potential therapeutic avenues [14].

In fact, the analysis of the literature on the topic in the last years highlighted the existence of a tight interplay between metabolic therapies largely used in the management of dysmetabolism, and ncRNA modulation. The evidence collected so far highlights their collective impact on cardiovascular fibrosis through the modulation of specific classes of ncRNAs. Also, a potential use of ncRNAs as biomarkers can be envisioned, together with the translational implications of the pharmacological interaction with the described molecular mechanisms deranged in dysmetabolic diseases mediating cardiovascular fibrosis. 

## 2. Metabolic Modulators and Their Influence on ncRNAs

### 2.1. Metformin

Metformin is a first-line treatment for type 2 diabetes. It activates AMP-activated protein kinase (AMPK), leading to improved insulin sensitivity and reduced inflammation. This same drug has shown promise beyond glycemic control by modulating fibroblast senescence via ncRNAs [15]. Recent evidence underscores metformin profound influence on epigenetic modulation via ncRNAs, notably miRNAs and lncRNAs, which are central players in myocardial fibrosis and diabetic cardiomyopathy. In fact, metformin not only affects metabolic pathways like AMPK and transforming growth factor-beta (TGF-β) signaling but also remodels post-transcriptional gene regulation, altering the expression of fibrotic genes via ncRNA networks. This modulation impacts fibroblast activation, ECM deposition, cardiac remodeling, and apoptosis. Metformin influences myocardial fibrosis not only via direct metabolic effects but also through epigenetic regulation of key miRNAs (e.g., miR-29, miR-133, miR-1) and lncRNAs (e.g., H19, nuclear paraspeckle assembly transcript 1 (NEAT1), metastasis associated lung adenocarcinoma transcript 1 (MALAT1), antisense of IGF2R non-protein coding RNA (AIRN)). These RNAs govern fibroblast behavior, apoptosis, and ECM accumulation. This growing body of work supports repositioning of metformin as a potential epigenetic modulator in cardiac therapeutics.

Investigations on diabetic rat hearts found that metformin treatment led to significant downregulation of miR-21, a miRNA widely implicated in promoting fibrosis by enhancing TGF-β signaling. The study demonstrated that reducing miR-21 corresponded with an upregulation of Suppression of Mothers Against Decapentaplegic Homolog 7 (Smad7), an inhibitory Smad that attenuates TGF-β-induced profibrotic responses [16]. This mechanistic insight supports the notion that metformin anti-fibrotic effects occur through ncRNA-mediated modulation of key signaling pathways. Complementary studies in skeletal muscle cells showed that metformin suppressed miR-21 in a concentration-dependent manner, correlating with improved insulin sensitivity, further suggesting systemic modulation of this miRNA across tissues [17].

In addition to miR-21, metformin was found to modulate other regulatory axes. A recent study in cardiac fibroblasts from diabetic murine models (db/db mice) revealed that metformin upregulated gene associated with retinoid-IFN-induced mortality-19 (Grim-19) and sirtuin 1 (SIRT1) while inhibiting phosphorylation of signal transducer and activator of transcription 3 (STAT3), resulting in decreased fibroblast proliferation and migration in high-glucose environments [18]. Grim-19 is known to participate in mitochondrial electron transport and cellular stress responses, while SIRT1 deacetylates multiple transcription factors involved in fibrosis. This multi-level regulation highlights how metformin impacts both ncRNA expression and protein signaling pathways.

Interestingly, a recent study highlighted that metformin induces the mitochondrial unfolded protein response (UPR_mt) via activation of heat shock factor 1 (HSF1). Activation of UPR_mt improves mitochondrial quality control and reduces oxidative stress, thereby protecting against hypertensive cardiac remodeling. Although direct ncRNA interactions were not assessed in this study, the pathways involved are known to interact closely with miRNA networks, suggesting an additional indirect mechanism of fibrosis [19].

A few analyses on large scale datasets were recently published showing the links between diabetes and its treatment and ncRNA dysregulation. Of particular interest were two studies depicting the diabetic mouse mRNA/ncRNA landscape and possible treatment options based on metformin.

The first study, from Zhao et al., investigated the differential expression of lncRNAs and mRNAs in the myocardium of diabetic cardiomyopathy (DCM) mice to explore the molecular mechanisms behind DCM pathogenesis. The study collected data from high-throughput RNA sequencing of heart tissues, echocardiography, and histology analyses of db/db mice compared to healthy controls. LncRNA–mRNA co-expression networks were constructed and revealed 93 lncRNAs and 881 mRNAs were significantly dysregulated in db/db mouse hearts. Dysregulated lncRNAs were linked with myocardial fibrosis, hypertrophy, and cardiomyocyte apoptosis, and Gene Ontology / Kyoto Encyclopedia of Genes and Genomes (GO/KEGG) analysis implicated immune signaling, metabolic processes, and apoptosis-related pathways. This paper offered the first comprehensive lncRNA-mRNA co-expression map in DCM, suggesting that specific lncRNAs may serve as biomarkers or therapeutic targets. While not directly testing metformin, the data provide a foundational landscape for studies evaluating therapeutic modulation of these lncRNAs [20].

The second study from Safavi et al. was designed to evaluate the combined effect of *Hippophae rhamnoides* L. (sea buckthorn, SBU) extract and metformin on DCM in type-2 diabetic (T2DM) mice, with a focus on genetic regulation, especially lncRNAs, and the ferroptosis pathway. Bioinformatic analysis of datasets obtained identified key genes (SERPINE1, EGFR, PTGS2, and CCL2) and biomarkers (lncRNAs NEAT1 and MALAT1) associated with inflammation, apoptosis, and fibrosis, supported by protein-protein interaction networks and in silico molecular docking to evaluate compound-target binding. This study demonstrated that the combination of sea buckthorn (SBU) and metformin has a synergistic effect in improving insulin sensitivity, reducing inflammatory cytokines like oncostatin, and reversing fibrosis in DCM. The therapeutic benefits were linked to the downregulation of lncRNAs NEAT1 and MALAT1 and to the modulation of 18 key hub genes involved in ferroptosis and inflammation, and highlighted the clinical potential of a dual-target strategy combining anti-fibrotic and lncRNA-based interventions [21]. The evidence collected from studies on lncRNA and organ fibrosis revealed a central role of MALAT-1 in the development of the condition. In the heart MALAT-1 appeared to exert its function as inflammation booster or directly mediating collagen gene expression [22,23]. This lncRNA, is upregulated in the peripheral blood of T2DM patients [24] and was found to be downregulated in the circulation of T2DM patients treated with metformin [25]. Given the established role of MALAT1 in cardiac fibrosis establishment and progression [22,23], further evidence would be needed to specifically address the effects of Metformin on the expression of this lncRNA. 

Finally, a comprehensive review explored how metformin, beyond its glucose-lowering effects, modulates epigenetic mechanisms, particularly through miRNAs and lncRNAs, to exert anti-inflammatory, anti-fibrotic, and vascular protective effects in diabetes and related conditions. The review highlighted key regulatory axes involving AMPK activation, DNA methylation, histone modifications, and RNA-mediated gene expression, emphasizing the potential of metformin as an epigenetic modulator in the prevention and treatment of diabetic complications [26].

The discussed evidence is collected in Table 1, highlighting key aspects of metformin interaction with ncRNA expression and function.

### 2.2. SGLT2 Inhibitors

Sodium-glucose cotransporter 2 (SGLT2) inhibitors, including empagliflozin, dapagliflozin, and canagliflozin, have revolutionized the treatment landscape for patients with diabetes and heart failure. Initially developed to improve glycemic control by inhibiting renal glucose reabsorption, these agents have demonstrated to have remarkable cardiovascular benefits beyond their metabolic effects [27,28]. Increasing evidence now implicates their role in modulating myocardial remodeling, particularly cardiovascular fibrosis, through intricate regulation of ncRNAs, especially miRNAs.

Among these, empagliflozin has gathered considerable attention for its ability to attenuate fibrosis via downregulation of pro-fibrotic miRNAs. One of the most studied targets is miR-21, a miRNA extensively linked to fibrotic and inflammatory signaling in the heart. MiR-21 promotes activation of cardiac fibroblasts and excessive ECM production by targeting key negative regulators of the TGF-β signaling pathway, including Smad7 and phosphatase and tensin homolog (PTEN) [29,30]. Ridwan et al. demonstrated that empagliflozin administration in diabetic mice led to significant downregulation of miR-21 [28]. This evidence, accompanied by diminished activation of the TGF-β/Smad pathway and lower levels of fibrotic markers such as fibronectin and α-smooth muscle actin (α-SMA) observed in diabetic models treated with the drug [31,32], supported a link between SGLT2 inhibition, miR-21 modulation, and anti-fibrotic outcomes. 

A recent analysis on miRNA expression in nicotinamide/streptozotocin-induced T2DM rats showed empagliflozin treatment increased the expression of miR-146a and miR-34a, two cardioprotective miRNAs downregulated in diabetic animals, suggesting a potential cardioprotective effect of this molecule exerted through miRNAs [33].

Additional clinical observations in patients with heart failure with preserved ejection fraction (HFpEF) and diabetes have reported similar reductions in circulating miR-21 and miR-92 following empagliflozin treatment, further corroborating the translational relevance of these findings. MiR-92, implicated in the fibrotic signaling, may act in concert with miR-21 to promote maladaptive remodeling, highlighting the multifaceted influence of SGLT2 inhibitors on cardiac miRNA profiles [34].

Beyond miR-21, SGLT2 inhibitors have also been shown to upregulate the miR-29 family widely recognized for its anti-fibrotic properties [35]. MiR-29 targets a broad range of ECM genes, including collagens, fibrillin, and elastin, thereby limiting the excessive matrix accumulation that underpins myocardial stiffening and fibrosis [36]. Restoration of miR-29 expression by empagliflozin in diabetic heart models is proposed to contribute substantially to the observed attenuation of cardiac fibrosis and improved ventricular compliance [37].

A particularly intriguing aspect of SGLT2 inhibitor action involves the regulation of SGLT2 expression itself by miRNAs, exemplified by miR-141. Studies have revealed that myocardial infarction induces downregulation of miR-141 in rat hearts. Conversely, overexpression of miR-141 in cultured cardiac fibroblasts suppresses SGLT2 expression, which in turn attenuates TGF-β–induced fibroblast proliferation and collagen synthesis. This finding suggests a feedback mechanism whereby miRNAs can both regulate and be regulated by SGLT2, creating a complex network that modulates fibrotic signaling pathways. The interplay between miR-141 and SGLT2 provides a mechanistic link bridging metabolic regulation with fibrotic responses in cardiac tissue [38].

Overall, the accumulating data suggest that the cardioprotective effects of SGLT2 inhibitors are mediated, at least in part, by their capacity to modulate miRNA expression profiles both within the myocardium and systemically. This dual modulation, through downregulation of pro-fibrotic miRNAs like miR-21 and miR-92 and upregulation of anti-fibrotic miRNAs such as miR-29, supports a novel, multifaceted mechanism for the anti-fibrotic properties of these drugs. The miRNA-mediated regulation of the TGF-β pathway and ECM synthesis positions SGLT2 inhibitors as promising therapeutic agents in the management of cardiovascular fibrosis associated with diabetes and heart failure. Further research on the precise miRNA networks and exosomal signaling pathways influenced by these drugs will likely yield deeper insights into their mechanism of action and potentially reveal new therapeutic targets.

The discussed evidence is collected in Table 2, highlighting key aspects of SGLT2 inhibitors on ncRNA expression and function.

### 2.3. PPARγ Agonists (Pioglitazone and Rosiglitazone)

Peroxisome proliferator-activated receptor gamma (PPARγ) agonists, like pioglitazone, are known for their insulin-sensitizing effects. In pressure overload-induced cardiac hypertrophy models, pioglitazone alleviated cardiac fibrosis and inhibited endothelial-to-mesenchymal transition (EndMT). While direct links to specific ncRNAs in this context are limited, PPARγ activation has been associated with modulation of miRNAs involved in inflammation and fibrosis.

Recent studies have shed light on the regulatory role of PPARγ activation in modulating miRNAs involved in inflammation and myocardial fibrosis. In diabetic cardiomyopathy models, PPARγ agonists such as pioglitazone and rosiglitazone have been shown to downregulate TGF-β/ERK signaling and inflammatory cytokines, resulting in reduced myocardial fibrosis, possibly via miRNA-mediated control of epithelial-to-mesenchymal transition pathways [39]. PPARγ activation has been shown to upregulate miR-711, which is a suppressor of specificity protein 1 (SP1) and downstream collagen-I expression, indicating a direct anti-fibrotic mechanism in myocardial infarction [40]. Furthermore, it has been reported that targeting miR-199b, a fibrosis-promoting miRNA elevated when PPARγ is suppressed, alleviated myocardial fibrosis in experimental models [35]. Other works have identified miR-29 and miR-133a as key anti-fibrotic miRNAs whose expression is enhanced through PPARγ signaling [41,42]. These findings collectively establish that PPARγ activation orchestrates a protective anti-fibrotic and anti-inflammatory network via miRNA and lncRNA regulation, making it a promising therapeutic target in myocardial fibrosis. Also, a combined strategy involving miR-29 antisense molecules and PPARγ agonists has been reported to show promising cardioprotective effects [43].

Additionally, Zhuang et al. have revealed that lncRNA, which increased in murine infarcted hearts and in activated cardiac fibroblasts, can inhibit the expression of PPARγ and its anti-fibrotic function. Rosiglitazone abolishes the pro-fibrotic effects of lncR-30245 in cardiac fibroblasts, specifically reducing the expression of Connective Tissue Growth Factor (CTGF), collagen I, and collagen III [44].

Finally, the lncRNA MALAT1 was described to enhance PPARγ methylation through Enhancer Of Zeste Homolog 2 (EZH2) binding, thereby reducing its anti-inflammatory action and exacerbating post-infarct inflammation and fibrosis [38]. The study of this molecular effect could possibly be expanded to cardiac fibrosis linked to metabolic diseases. The expression of MALAT1, increased in T2DM [24] could indeed limit the efficacy of PPARγ agonists.in this pathological context. 

### 2.4. GLP-1 Receptor Agonists (Liraglutide)

Glucagon-like peptide-1 (GLP-1) receptor agonists, such as liraglutide, have shown promise in reducing oxidative stress and apoptosis in cardiac tissues, and have demonstrated cardioprotective effects that extend beyond glucose lowering, notably via ncRNA modulation [45]. A key focus has been on the miR-21 and miR-200 family (including miR-200a, miR-200b, miR-200c, miR-429, and miR-141), which play a critical role in controlling several aspects of fibrotic activation such as epithelial-to-mesenchymal transition, TGF-β and PI3K signaling, and ECM deposition, all of which are central to the pathogenesis of myocardial fibrosis. Studies have shown that GLP-1 agonist Liraglutide influence miR-21 expression in atrial fibroblasts treated with Angiotensin II. Liraglutide administered in vitro could reduce the fibroblasts activation phenotype, measured as proliferative and invasive capacity. The molecular mechanism included the down regulation of miR-21, induced by the Angiotensin stimulus. This recent evidence shows a direct effect of Liraglutide on miRNA expression, mediating fibrotic commitment of primary cardiac fibroblasts [46]. Several lines of evidence have been collected on the role of miR-200 family in diabetes. The expression of miR-200b and miR-200c is increased by GLP1 in the liver, and these miRNAs reduce intra hepatic lipid accumulation and liver steatosis [47]. The level of miR-200 is regulated by a complex epigenetic circuitry acting also in the cardiovascular system. Studies in tissues affected by metabolic diseases have shown that enhanced miR-200 expression has an antifibrotic effect, and the miRNA was found to be downregulated in fibrotic renal and pulmonary tissues [48]. Pharmacological treatment with Sitagliptin, a DPP-4 inhibitor which mimics the function of GLP1 agonists, was shown to restore miR-200b and miR-200c expression [49]. Suppression of miR-200b leads to cardiac fibroblast autophagy and cardiac fibrosis through DNMT3A [50], and miR-200 has been recently described to have a protective effect in cardiac fibrosis. There is no evidence yet on the effects of GLP1 agonists on miR-200 in the heart, but the evidence collected in other organs affected by metabolic conditions would suggest a beneficial action also in the myocardium. Together, these findings underscore a biologically significant interplay between GLP-1 receptor agonists and two miRNAs with a prominent role in cardiac fibrosis development such as miR-21 and miR-200. Pharmacological modulation as well as administration through RNA based drugs could be a new weapon for the control of myocardial fibrotic remodeling.

### 2.5. Fatty Acid Oxidation Inhibitors (Trimetazidine)

Trimetazidine (TMZ), an inhibitor of fatty acid oxidation, has been explored for its cardioprotective effects through ncRNA modulation. Several recent studies have explored the role of TMZ in modulating miR-133a, a cardioprotective miRNA known for its anti-fibrotic and anti-apoptotic effects in the myocardium. TMZ primary metabolic action is the inhibition of long-chain 3-ketoacyl-CoA thiolase, leading to a shift from fatty acid oxidation to glucose utilization in cardiac cells, which improves energy efficiency under ischemic or diabetic conditions. Ghosh et al. have reported that TMZ treatment in diabetic cardiomyopathy models upregulated miR-133a, alongside miR-1 and miR-499, resulting in reduced myocardial fibrosis, oxidative stress, and apoptosis [51]. Similarly, overexpression of miR-133a alleviates ERK1/2-mediated fibrotic signaling in diabetic hearts, a pathway also targeted by TMZ [42,52]. Another study by Pan et al. has found that miR-133a-3p directly inhibits the inhibitor of nuclear factor kappa-B kinase ε (IKKε) and suppresses pyroptosis, a form of programmed inflammatory cell death [53,54], offering further insight into TMZ anti-inflammatory cardiac effects. These findings collectively position TMZ as a promising therapeutic agent for cardiac remodeling and fibrosis by restoring miR-133a levels, which in turn regulates fibrotic gene expression, improves mitochondrial metabolism, and attenuates cardiomyocyte death.

Moreover, in hypertensive rat models, TMZ has been shown to increase miR-30a expression, which suppressed connective tissue growth factor (CTGF), a potent mediator of cardiac fibrosis. This change has improved cardiac compliance and reduced pathological ECM remodeling [55]. Together, these data highlight the capacity of fatty acid oxidation inhibitors to modulate key anti-fibrotic ncRNAs and support their potential as adjunctive therapies in cardiovascular fibrosis.

### 2.6. Cardiometabolic Therapies and CircularRNAs: A Current Knowledge Gap

Despite compelling evidence links specific circRNAs to cardiac fibrosis, studies that directly evaluate how established cardiometabolic therapies modulate circRNA expression in cardiac tissue or cardiac fibroblasts, remain scarce. For example, circNFIB and circIGF1R have been shown to attenuate cardiac fibroblast activation and extracellular matrix accumulation, implicating circRNA-mediated networks in antifibrotic remodeling [56,57]. Several circRNAs have been identified as pivotal regulators of fibrotic remodeling, including circ_000203 [58], circSMAD4 [59], and circPVT1 [60]. Current preclinical and transcriptomic data demonstrate that cardiometabolic therapies alter gene expression, reduce fibrosis markers, and improve cardiac structure/function, yet without parallel measurements of circRNAs.

Pharmacological studies outside the heart indicate that drugs such as dapagliflozin and metformin can profoundly reshape circRNA expression. For example, in proximal tubular cells, these agents normalized hyperglycemia-induced changes, including regulation of hsa_circRNA_012448 via the miR-29b-2-5p/GSK3β axis [61,62]. Similarly, circACC1 has been identified as a key activator of AMPK signaling, a canonical target of metformin, establishing a mechanistic bridge between circRNAs and metabolic stress responses [63]. Furthermore, hsa_circ_0072309 was shown to engage the PPARγ/PTEN pathway, with its effects potentiated by the PPARγ agonist pioglitazone [64]. While these findings highlight mechanistic plausibility for circRNA involvement in the antifibrotic actions of cardiometabolic drugs, direct evidence in the heart is currently lacking. Future studies should specifically address whether circRNA modulation by metabolic modulators constitutes a novel therapeutic mechanism and biomarker axis linking metabolic interventions to improved cardiac fibrosis outcomes.

## 3. Therapeutic Implications: The Interplay Between Metabolic Modulators and ncRNAs Offers Novel Therapeutic Strategies and Markers of Treatment Efficacy for Cardiovascular Fibrosis

The interplay between metabolic modulators and ncRNAs represents an emerging therapeutic axis in the context of cardiovascular fibrosis. Cardiac fibroblasts undergo profound metabolic reprogramming during activation, shifting toward glycolysis and altered mitochondrial function to support ECM production and fibrotic remodeling. Concurrently, ncRNAs, including miRNAs, lncRNAs, and circRNAs, have been identified as critical regulators of both fibroblast phenotype and cardiac metabolism [2]. This bidirectional relationship enables ncRNAs to modulate key metabolic pathways, while metabolic cues can, in turn, influence ncRNA expression and function. 

Therapeutically, this crosstalk offers a novel opportunity for precision intervention: metabolic drugs such as PPAR agonists or AMPK activators can reshape fibroblast metabolism [65,66,67], and the design of ncRNA-based strategies can fine-tune pro-fibrotic gene expression networks. Recent studies suggest that combinatorial targeting of metabolic and ncRNA pathways may yield synergistic effects in reversing fibroblast activation and attenuating fibrosis [68]. Leveraging this integrative approach could pave the way for more effective and specific anti-fibrotic therapies in cardiovascular disease. Although most extensively studied in renal tissue, a recent study combining fenofibrate (a PPARα agonist) with miR-668 mimics demonstrated superior suppression of fibrosis compared to either intervention alone, suggesting that similar synergistic strategies may be translated to cardiac models where PPARα and mitochondrial metabolism are impaired. Collectively, these findings highlight the therapeutic promise of integrating metabolic reprogramming and ncRNA modulation in cardiovascular fibrosis. Such dual-targeting approaches may offer enhanced efficacy by simultaneously disrupting upstream metabolic drivers and downstream gene expression programs that sustain fibroblast activation and matrix accumulation.

Moreover, ncRNAs are increasingly recognized as dynamic biomarkers that reflect disease state and therapeutic response. Metabolic therapies, such as AMPK activators, PPAR agonists, and mitochondrial-targeting drugs, influence cellular energy balance and redox status, which are tightly linked to ncRNA expression. As these therapies modulate signaling networks and metabolic fluxes, they often induce distinct changes in the expression profile of ncRNAs involved in fibrosis, inflammation, and cell survival. As presented in our review of the recent literature, several miRNAs, such as miR-21, a well-known pro-fibrotic miRNA, miR-29, a key anti-fibrotic miRNA, and the lncRNA MALAT1, which plays a dual role in metabolic and fibrotic signaling, have been reported to be responsive to metabolic interventions. Similarly, PPARγ agonists such as pioglitazone have been shown to alter circulating levels of several miRNAs [69]. Given the possibility for these ncRNAs to be measured in blood and other accessible biofluids, these therapy-responsive ncRNAs hold promise as minimally invasive biomarkers for monitoring treatment efficacy in cardiovascular and fibrotic diseases which develop in metabolic disease conditions. Importantly, their dynamic expression in response to metabolic cues reflects both the metabolic state of target cells (e.g., fibroblasts, endothelial cells) and the success of therapeutic modulation, providing a functional readout that integrates molecular, cellular, and systemic responses. 

Recent clinical evidence supports the utility of circulating and exosomal ncRNAs as biomarkers to monitor the efficacy of cardiometabolic therapies. In a pilot study of patients with type 2 diabetes, distinct circulating miRNA signatures were associated with clinical response to GLP-1 receptor agonist therapy, highlighting their predictive potential for treatment stratification [70]. A more recent review confirmed that GLP-1RAs, either alone or in combination with metformin, consistently modulate both exosomal and non-exosomal circulating miRNAs, underscoring their role as dynamic biomarkers of therapeutic response [71]. In the context of metformin therapy, reductions in circulating miR-19a and miR-221 were observed in patients with type 2 diabetes and myocardial infarction, linking these pro-fibrotic and pro-atherogenic miRNAs to the cardioprotective effects of the drug [72]. Similarly, in the TODAY clinical trial, elevated levels of miR-483-3p and miR-4306 were associated with treatment failure in youth with type 2 diabetes, further supporting the prognostic role of circulating miRNAs in therapy monitoring [73]. Collectively, these studies emphasize that specific ncRNAs may serve not only as mechanistic mediators but also as clinically relevant biomarkers to evaluate cardiometabolic drug efficacy.

Circulating microRNAs are attractive biomarkers due to their stability in biofluids and their disease-related expression patterns [74,75]. The studies mentioned above include evidence of miRNAs specifically linked to fibrotic complications in the cardiovascular system that could allow monitoring of metabolic modulator drugs treatment progression [72]. Nevertheless, their clinical translation is limited by considerable technical variability and lack of methodological standardization across studies [76,77,78]. This variability arises at the pre-analytical, analytical, and post-analytical levels, and strongly impacts on the comparability of results [78]. Pre-analytical factors are the single largest source of variation for circulating miRNA measurements. Choice of biofluid (plasma vs serum vs whole blood), blood-collection tube (EDTA, citrate, serum separator, preservative tubes), time to processing, centrifugation speed and number of spins, storage temperature, and freeze–thaw cycles all change the measured miRNA profile—sometimes dramatically for specific miRNAs [74,76,79,80,81]. Studies directly comparing protocols show that centrifugation regime and tube type can increase apparent levels of certain miRNAs or affect hemolysis, which in turn confounds measurements [81]. Different detection platforms introduce additional noise. RT-qPCR remains the standard for validation, even though it is still subject to varying levels of primer specificity and amplification efficiency [82]. Small RNA sequencing enables discovery but suffers from ligation and library preparation biases, while bioinformatic pipelines can yield divergent results from identical datasets [83]. Beyond technical noise, biological factors (age, sex, circadian rhythm, diet, smoking, medication, co-morbidities, pregnancy) and within-person longitudinal variability influence circulating miRNA levels [80,84,85]. Some miRNAs are remarkably stable intra-individually, but others are sensitive to lifestyle [86,87] or transient physiological states [88]. Accounting for these covariates during study design and analysis (matched controls, covariate adjustment) is crucial for finding disease-specific signals. In summary, the promise of circulating miRNAs as biomarkers lies not only in discovery alone but in controlling and documenting technical variability across the full workflow. Adoption of standardized protocols, rigorous QC, and transparent reporting will be decisive for their reproducibility and eventual clinical implementation.

Finally, extracellular vesicles, and in particular exosomes (30–150 nm), have emerged as crucial mediators of intercellular communication by transporting proteins, lipids, and ncRNAs. Importantly, the exosomal ncRNA profile reflects the physiological or pathological state of the parent cell, suggesting that cardiometabolic therapies may not only regulate ncRNA expression within cells but also alter their extracellular packaging and intercellular transfer. Although still limited, emerging studies suggest that cardiometabolic therapies can influence the ncRNA cargo of extracellular vesicles, thereby shaping intercellular signaling in metabolic and cardiovascular disease. Diabetes and metabolic stress are associated with shifts in exosomal miRNA composition, and therapeutic interventions may harness this mechanism both as biomarkers and as mediators of antifibrotic effects [89]. In preclinical and clinical studies, GLP-1 receptor agonists (alone or in combination with metformin) were shown to modulate circulating exosomal and non-exosomal miRNAs associated with glucose metabolism and cardiovascular health [71]. The therapeutic approaches mediated by exosomal delivery of RNA has been tested in pathological models different from the cardiovascular disease but were demonstrated to act directly on fibroblasts in the targeted organs. As an example of therapeutic approaches translatable to the cardiovascular system, exosomes enriched in antifibrotic miRNAs such as miR-29 have been shown to attenuate extracellular matrix deposition in skin scar formation and improve tissue regeneration in preclinical models [90], while engineered exosomes carrying siRNA against TGF-β1 signaling activation successfully reduced liver profibrotic remodeling in vivo [91]. Given that metabolic modulators such as metformin, SGLT2 inhibitors, GLP-1 receptor agonists, and PPARγ agonists converge on similar ncRNA-TGF-β pathways, it is plausible that part of their action may involve modifying the exosomal ncRNA cargo released from cardiac and/or vascular cells. From a translational perspective, this raises two major implications: circulating exosomal ncRNAs may serve as dynamic biomarkers reflecting the efficacy of cardiometabolic therapies on fibrotic remodeling, and engineered exosomes could be exploited as therapeutic vectors to deliver antifibrotic ncRNAs directly to diseased myocardium, offering a novel strategy to complement metabolic interventions.

## 4. Conclusions

Metabolic modulators influence the expression of ncRNAs involved in cardiovascular fibrosis, highlighting a convergence between metabolic regulation and epigenetic control. This crosstalk suggests that metabolic status can directly impact gene expression patterns not only through canonical pathways like energy sensing and redox balance but also by modulating ncRNA networks that govern fibroblast activation, ECM remodeling, and inflammatory signaling. A growing body of literature shows the ability of several metabolic modulators to tune the expression of miRNAs and lncRNAs implicated in fibrosis-related signaling cascades (Figure 1). Understanding these interactions opens new therapeutic avenues, where targeting metabolic pathways may indirectly reshape the epigenetic landscape of fibrotic tissues. Moreover, the few synergistic approaches examined, still at the preclinical level, suggest combinatorial therapeutic interventions may significantly improve the efficacy of cardiometabolic treatments. Also, several ncRNAs emerge from multiple studies as not only key effectors in fibrotic remodeling but also as therapy-responsive circulating biomarkers. This highlights a promising translational avenue for the use of ncRNAs as minimally invasive, dynamic biomarkers. 

Across the drug classes examined, it becomes evident that while all cardiometabolic therapies converge on attenuating fibrotic remodeling, they do so via distinct ncRNA networks. Metformin and SGLT2 inhibitors share the ability to suppress pro-fibrotic miR-21, thereby dampening TGF-β/Smad signaling, whereas PPARγ agonists preferentially upregulate anti-fibrotic miRNAs such as miR-29 and miR-133a, reinforcing extracellular matrix regulation. GLP-1 receptor agonists target miR-21, directly linking their action to the signaling pathways which activates fibroblasts, while fatty acid oxidation inhibitors such as trimetazidine restore cardioprotective miR-133a levels and reduce pyroptosis. These differences highlight drug-class specific ncRNA signatures that may allow the design of combinatorial approaches. This also highlights that ncRNA regulation not only mediates shared antifibrotic effects but also distinguishes the mechanistic profiles of each therapy.

Despite the recent evidence collected, and the possibility to use miRNAs as biomarkers for disease conditions as well as predictors of clinical outcome of ongoing therapies, several barriers remain before ncRNAs can be fully integrated into routine clinical practice as reliable biomarkers. These include the lack of assay standardization, variability in RNA quantification methods (e.g., qPCR, RNA-seq, microarrays), and heterogeneity in patient cohorts, which limit reproducibility and comparability across studies. These limitations reduce confidence in the predictive power of ncRNA signatures and underscore the need for systematic, longitudinal, and multicenter validation studies. 

The feasibility of new drug design to use miRNA as coadjuvants or possibly substitute for current treatments for metabolic diseases, is still limited by the available data. In fact, while many ncRNAs have been associated with disease endpoints, causal relationships are not always consistently demonstrated with rigorous functional validation.

Critically, while the current review highlights correlations between drug classes and ncRNA modulation, several studies rely on small animal models or in vitro data, and the field suffers from publication bias favoring positive results, thus giving a partial picture of the molecular aspects covered. 

Further research on these mechanisms should move beyond descriptive studies and invest in validation pipelines that prioritize reproducibility, functional testing, and clinical feasibility. Looking ahead, the therapeutic manipulation of ncRNAs presents several exciting possibilities, including the development of ncRNA-based therapeutics such as miRNA mimics or inhibitors, especially for well-characterized targets, and the test of combinatorial strategies, pairing metabolic modulators with targeted ncRNA therapies. 

In conclusion, the integration of ncRNA biology into the study of metabolic therapies for cardiac fibrosis offers exciting potential for precision medicine approaches. However, this field remains in its early translational phase. Future investigations in this field will enhance our ability to develop precision therapies that exploit the interplay between metabolism and ncRNA regulation to halt or even reverse cardiovascular fibrosis.

## Figures and Tables

**Figure 1 metabolites-15-00664-f001:**
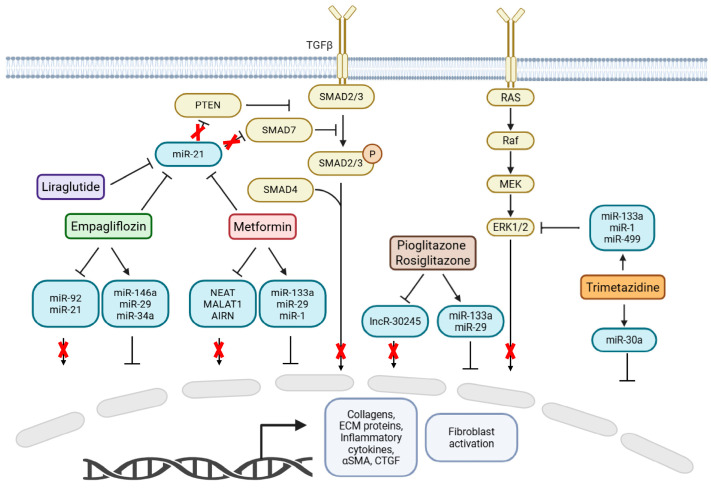
Schematic overview of the main metabolic modulator drug-ncRNA-signaling pathway axes involved in the regulation of cardiovascular fibrosis. Scheme of the main pathways (yellow boxes) that mediate the fibrotic process activation (gray boxes), and the ncRNAs (blue boxes) involved in the regulation of these signaling pathways. The effect of different classes of cardiometabolic drugs on ncRNAs is shown (red box: Metformin; green box: Empagliflozin, SGLT2 inhibitor; purple box: Sitagliptin, inhibitor of the GLP-1 peptidase; brown box: Pioglitazone and Rosiglitazone, PPARγ agonists; orange box: Trimetazidine, fatty acid oxidation inhibitor).

**Table 1 metabolites-15-00664-t001:** Summary table for effects on cardiovascular fibrosis via non-coding RNAs of Metformin.

Metformin	Details	References	Study Context
**Primary** **Action**	Activates AMPK, improves insulin sensitivity, reduces inflammation	[15]	In vivo and in vitro
**Epigenetic** **Mechanism**	Modulates non-coding RNAs (miRNAs and lncRNAs), affects post-transcriptional gene regulation of fibrotic genes	[15,26]	In vivo and in vitro
**Key miRNAs** **Affected**	miR-21 (downregulated), miR-29, miR-133, miR-1	[16,17]	In vivo and in vitro
**Key lncRNAs** **Affected**	H19, NEAT1, MALAT1, AIRN	[21,26]	In vivo and in silico
**Fibrotic Pathways** **Modulated**	TGF-β signaling via downregulation of miR-21 and upregulation of Smad7 (inhibitory Smad)	[16]	In vitro
**Effects on** **Fibroblasts**	Decreased proliferation and migration; linked to Grim-19 and SIRT1 upregulation; STAT3 phosphorylation inhibited	[18]	In vivo and in vitro
**Mitochondrial** **Effects**	Induces mitochondrial unfolded protein response (UPR_mt) via HSF1, improving mitochondrial quality and reducing oxidative stress	[19]	In vivo and in vitro
**Animal** **Models**	Diabetic rat hearts, db/db mouse cardiac fibroblasts	[16,18,20]	In vivo and in vitro
**High-throughput** **Studies**	Identified dysregulated lncRNAs and mRNAs in diabetic cardiomyopathy (93 lncRNAs, 881 mRNAs); linked to fibrosis and apoptosis	[20]	In vivo
**Combination** **Therapies**	Synergistic effect with Hippophae rhamnoides L. (sea buckthorn) reducing fibrosis, inflammation via downregulating NEAT1 and MALAT1	[21]	In vivo and in silico
**Clinical** **Potential**	Repositioning as epigenetic modulator in cardiac therapeutics targeting ncRNA networks to reduce fibrosis and inflammation	[15,26]	In vivo and in vitro

**Table 2 metabolites-15-00664-t002:** Summary table for modulation of non-coding RNAs in cardiovascular fibrosis of SGLT2 Inhibitors.

SGL2 Inhibitors	Details	References	Study Context
**Primary** **Action**	Inhibit renal glucose reabsorption, improve glycemic control	[21,22]	In vivo and in silico
**Cardiovascular** **Benefits**	Beyond metabolism: reduce myocardial remodeling and cardiac fibrosis via ncRNA regulation	[21,22]	In vivo and in silico
**Key miRNAs** **Downregulated**	miR-21 (pro-fibrotic, promotes fibroblast activation via TGF-β pathway by targeting Smad7 and PTEN), miR-92	[22,23,24,28]	In vivo, in vitro and in silico
**Key miRNAs** **Upregulated**	miR-29 (antifibrotic, targets ECM genes such as collagens, elastin), miR-146a, miR-34a	[27,29,30]	In vivo and in vitro
**Mechanistic** **Insights**	Downregulation of miR-21 linked to decreased TGF- β/Smad pathway activity and reduced fibrosis markers (fibronectin, α-SMA)	[22,25,26]	In vivo
**Animal** **Models**	Diabetic mice, nicotinamide/streptozotocin-induced T2DM rats	[22,27]	In vivo and in vitro
**Clinical** **Observations**	Reduced circulating miR-21 and miR-92 in HFpEF patients treated with empagliflozin	[28]	In vivo and in vitro
**miRNA-Drug** **Feedback Loop**	miR-141 downregulated after MI; overexpression suppresses SGLT2 in cardiac fibroblasts, reducing fibrosis via TGF-β pathway	[31]	In vivo and in vitro
**Overall** **Mechanism**	Cardioprotective effects via dual modulation of miRNAs: downregulating pro-fibrotic and upregulating anti-fibrotic miRNAs	[21,22,29]	In vivo, in vitro and in silico

## Data Availability

No new data were created or analyzed in this study.

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
