# Peer review of "Cardiometabolic Therapies Shape Non-Coding RNA Landscapes in Cardiovascular Fibrosis"

_metabolites, 2025, doi:10.3390/metabo15100664_

Round 1

Reviewer 1 Report

Comments and Suggestions for Authors

Please provide the search strategy in detail. Provide the information about databases, date ranges, search strings, language limits, grey literature sources etc.

Please provide a protocol, say for example PROSPERO/OSF. A PRISMA flow diagram of the study will improve the quality of the MS

Inclusion and exclusion criteria (population, models, interventions, comparators, outcomes, study designs) must be mentioned in detail

Pls mention in what way cardiac fibrosis and ncRNA modulation were operationalized, and how off-target tissue data (skeletal muscle, endothelium) were handled in the study

Pls provide information on the screening process (number of reviewers, training, inter-rater agreement/kappa), data-extraction templates, management of multiple reports from the same cohort, or in what means disagreements were resolved

I observed that there is no appraisal per study design (SYRCLE for animal studies, RoB 2 for RCTs, ROBINS-I for non-randomized, ARRIVE for preclinical reporting). Consequently, effect claims (empagliflozin reduces circulating miR-21/miR-91 in HFpEF with diabetes) are presented without grading the underlying evidence

I have to mention that animal, cell, and human data are intertwined without any stratification, and effectiveness claims (anti-fibrotic outcomes) are discussed alongside expression-only findings

There is a repeated mention of miR-91 as pro-fibrotic. Actually, this is unusual and may be a typographical error

Pls check ref.12. Inclusion of a future-dated reference (Dec 01, 2025) relative to the current date of this review. This may not have been available at the time of search. Pls cross-check and correct

There is a conflicting narrative on MALAT1 which is reported to enhance PPARγ methylation and exacerbate inflammation, yet else where pioglitazone is said to up-regulate MALAT1

Assertions about circulating ncRNAs as treatment-response biomarkers are not accompanied by inclusion criteria for assay type, pre-analytics, normalization, diagnostic performance, or validation stage

Comments on the Quality of English Language

There are a lot of grammatical errors and tense inconsistencies that have to be fixed

Author Response

Point-by-point response to Comments and Suggestions for Authors

  1. Please provide the search strategy in detail. Provide the information about databases, date ranges, search strings, language limits, grey literature sources etc.
    • We thank for the request. We have conducted a search on PubMed and set a filter for the last 10 years. We have added this information in the manuscript. Line 20: “A literature search was manually curated and conducted on PubMed for studies published mainly in the last decade evaluating the ….”

  1. Please provide a protocol, say for example PROSPERO/OSF. A PRISMA flow diagram of the study will improve the quality of the MS
    • We apologize for possibly not being clear with the type of bibliographical search performed. As stated above we manually curated the bibliographic search for relevant articles. Hence, the action requested does not apply to the type of review proposed, which is not a systematic review.

  1. Inclusion and exclusion criteria (population, models, interventions, comparators, outcomes, study designs) must be mentioned in detail
    • We believe this comment arises from the comment #2. The action requested does not apply to the type of review proposed, which is not a systematic review. We have selected the paper according to the publication date and the topic covered, without further filtering for other features of the data presented

  1. Pls mention in what way cardiac fibrosis and ncRNA modulation were operationalized, and how off-target tissue data (skeletal muscle, endothelium) were handled in the study
    • We believe this comment arises from the comment #3. The action requested does not apply to the type of review proposed, which is not a systematic review. We did read the abstracts and summary of single papers which came out of the search and mainly excluded those not related to cardiac tissue. We are possibly reporting here experimental evidence on other tissues affected by metabolic disorders, when a relevant link with fibrosis occurs or to anticipate possible strategies relevant for the treatment of cardiovascular fibrosis.
  2. Pls provide information on the screening process (number of reviewers, training, inter-rater agreement/kappa), data-extraction templates, management of multiple reports from the same cohort, or in what means disagreements were resolved
    • We believe this comment arises from the comment #3. The action requested does not apply to the type of review proposed, which is not a systematic review.

  1. I observed that there is no appraisal per study design (SYRCLE for animal studies, RoB 2 for RCTs, ROBINS-I for non-randomized, ARRIVE for preclinical reporting). Consequently, effect claims (empagliflozin reduces circulating miR-21/miR-91 in HFpEF with diabetes) are presented without grading the underlying evidence
    • The action requested does not apply to the type of review proposed, which is not a systematic review.

  1. I have to mention that animal, cell, and human data are intertwined without any stratification, and effectiveness claims (anti-fibrotic outcomes) are discussed alongside expression-only findings
    • The action requested does not apply to the type of review proposed, which is not a systematic review.
  2. There is a repeated mention of miR-91 as pro-fibrotic. Actually, this is unusual and may be a typographical error
    • We thank the reviewer for spotting the typo. We have corrected miR-91 to miR-92 throughout the manuscript as highlighted, according to the cited reference [34],  as well as in Table 2.
  3. Pls check ref.12. Inclusion of a future-dated reference (Dec 01, 2025) relative to the current date of this review. This may not have been available at the time of search. Pls cross-check and correct
    • We thank the reviewer for careful evaluation of the References section. We manually checked the entry in the software and corrected it. We have formatted all the references according to the journal style.

  1. There is a conflicting narrative on MALAT1 which is reported to enhance PPARγ methylation and exacerbate inflammation, yet else where pioglitazone is said to up-regulate MALAT1
    • We thank the reviewer for the insightful comment. We have checked the literature and found evidence about the role of MALAT1 in the regulation of inflammation. Based on this evidence, we proposed a potential expansion of the study of MALAT1 role in cardiac fibrosis linked to metabolic diseases as well. In addition, we reported a different lncRNA, lncR-30245, which is expressed in activated cardiac fibroblasts and is inhibited by the PPARγ agonist Rosiglitazone to reduce cardiac fibrosis.

Lines 256-266: Additionally, Zhuang et al. revealed that lncRNA lncR-30245, which is increased in murine infarcted hearts and in activated cardiac fibroblasts, inhibits the expression of PPARγ and its anti-fibrotic function. Rosiglitazone, a PPARγ agonist, abolishes the pro-fibrotic effects of lncR-30245 in cardiac fibroblasts, specifically reducing the expression of CTGF, collagen I, and collagen III

  1. Assertions about circulating ncRNAs as treatment-response biomarkers are not accompanied by inclusion criteria for assay type, pre-analytics, normalization, diagnostic performance, or validation stage
    • We believe this comment follows some listed above (comments # 2-7), applicable to the assessment of systematic reviews. The action requested does not apply to the type of review proposed, which is not a systematic review.

Comments on the Quality of English Language

There are a lot of grammatical errors and tense inconsistencies that have to be fixed.

  • We thank the reviewer for the kind suggestion. We have had the manuscript checked by a native English speaker, for extensive revision of English language use and corrected the manuscript accordingly.

Reviewer 2 Report

Comments and Suggestions for Authors

General Comment:
The review article presents a well-structured overview of biomarkers and cardiometabolic therapies in the context of cardiac fibrosis, supported by a thorough literature survey. The following suggestions are intended to enhance the clarity and depth of the content:

Comment 1:
The abstract mentions synergistic interventions in the conclusion; however, the main body of the article includes limited discussion and references on synergistic mechanisms that support therapeutic efficacy. Incorporating additional literature on various therapeutic combinations would strengthen this section.

Comment 2:
Adding an extra column to the tables indicating whether the data is derived from in-vitro, in-vivo, or human studies would improve clarity and help readers better interpret the translational relevance of the findings.

Author Response

We thank the reviewer very much for taking the time to review this manuscript and provide such positive feedback on its structure and content. Please find the detailed responses below and the corresponding revisions/corrections highlighted/in track changes in the re-submitted files

Point-by-point response to Comments and Suggestions for Authors

General Comment:
The review article presents a well-structured overview of biomarkers and cardiometabolic therapies in the context of cardiac fibrosis, supported by a thorough literature survey. The following suggestions are intended to enhance the clarity and depth of the content:

Comment 1:
The abstract mentions synergistic interventions in the conclusion; however, the main body of the article includes limited discussion and references on synergistic mechanisms that support therapeutic efficacy. Incorporating additional literature on various therapeutic combinations would strengthen this section.

  • We thank the reviewer for the comment on this aspect. We have indeed found a few examples of synergistic approaches, mainly performed in small animals and in translational research studies at a very initial stage. We yet propose that synergistic interventions taking advantage of both established drugs used in metabolic disease treatment and the modulation of ncRNAs is a future development for the establishment of new therapeutic approaches. We added this paragraph in the Conclusions section.

Lines: 452-454.“Moreover, the few synergistic approaches examined suggest combinatorial therapeutic interventions may significantly improve the efficacy of cardiometabolic treatments.”

Comment 2:
Adding an extra column to the tables indicating whether the data is derived from in-vitro, in-vivo, or human studies would improve clarity and help readers better interpret the translational relevance of the findings.

  • We thank the reviewer for providing help to increase clarity and usefulness of our work. We have added an extra column to the tables indicating whether the study was conducted in vitro, in vivo, or in silico. Please see the additional table heading “Study Context” in both Table 1 and Table 2.

Reviewer 3 Report

Comments and Suggestions for Authors

Type of manuscript: Review

Title: Cardiometabolic Therapies Shape Non-Coding RNA Landscapes in Cardiovascular Fibrosis; metabolites-3843783.

The manuscript  is a review on potential of  metabolic modulators  as therapeutic strategies and biomarkers in cardiovascular fibrosis. Results show that metabolic modulators exert anti-fibrituic effects by ncRNA regulation suggesting a novel therapheutic strategies and potential biomarkers for cardiovascular fibrosis.

Abstract

Authors must include the databases used in this review (line 20)

Line 26 change as  ……H19, and MALAT1)

Lines 14, 32 and all manuscript  the authors  must decide  the use of  cardiovascular fibrosis (title and line 340) or cardiac fibrosis.

Keywords

Change cardiac fibrosis by cardiovascular fibrosis; MicroRNAs as microRNAs; cardiometabolic diseases for cardiometabolic síndromes

Introduction

In all manuscript Check the format of refereneces in agreemment with the journal

Lines 43, 49, 52, 151,159,163,217,219,223,234,246,269,296,

Change [1],[2], [3] as [1-3]; [4],[5],[6] as [4-6]……………………

Line 78 change as  mir-133, and miR-1; MALAT1, and Airn

Line 124 L is normal as L

Line 127 change PTGS2, and CCL2

Line 247 delete (2021)

References

In all references the numbers are duplicated please delete the duplicate

References 1,2, 4, 7, 13-17,21,27,29,31,34,35,38,39,43,45-54 please include authors for et al

Reference 22, 43 use abbreviations of journal

Author Response

We thank the reviewer for thoroughly assessing the whole manuscript and provide helpful feedback on its structure and content. Please find the detailed responses below and the corresponding revisions/corrections highlighted/in track changes in the re-submitted files

Point-by-point response to Comments and Suggestions for Authors

Type of manuscript: Review

Title: Cardiometabolic Therapies Shape Non-Coding RNA Landscapes in Cardiovascular Fibrosis; metabolites-3843783.

The manuscript  is a review on potential of  metabolic modulators  as therapeutic strategies and biomarkers in cardiovascular fibrosis. Results show that metabolic modulators exert anti-fibrituic effects by ncRNA regulation suggesting a novel therapheutic strategies and potential biomarkers for cardiovascular fibrosis.

Abstract

  1. Authors must include the databases used in this review (line 20)
    • We thank for the suggestion and have add the detail in the text

Line 21:Methods: A literature search was manually curated and conducted on PubMed for studies published mainly in the last decade evaluating

  1. Line 26 change as  ……H19, and MALAT1)
    • We thank for the suggestion and have changed the text as follows.

Line 28: specifically downregulate pro-fibrotic (miR-21, miR-91, H19, and MALAT1) and upregulate…”

  1. Lines 14, 32 and all manuscript  the authors  must decide  the use of  cardiovascular fibrosis (title and line 340) or cardiac fibrosis.
    • We thank the reviewer for pointing this out. We have decided to keep the cardiovascular perspective as we discuss of the cardiovascular implications of the selected molecules and ncRNAs. Therefore, we have changed the expression “cardiovascular fibrosis” throughout the manuscript since we examine studies on both cardiac and vascular fibrosis. We have possibly been using the term “cardiac fibrosis” when referring to a specific study in which only myocardial fibrosis had been investigated.

Keywords

Change cardiac fibrosis by cardiovascular fibrosis; MicroRNAs as microRNAs; cardiometabolic diseases for cardiometabolic síndromes

  • We thank for the suggestions. We have changed the keywords section as follows

Line 38: “fibrotic remodeling; metabolic modulators; microRNAs; long non-coding RNAs; epigenetic regulation; cardiometabolic syndromes; therapeutic targets”

Introduction

  1. In all manuscript Check the format of refereneces in agreemment with the journal. Lines 43, 49, 52, 151,159,163,217,219,223,234,246,269,296, Change [1],[2], [3] as [1-3]; [4],[5],[6] as [4-6]………
    • We thank for the suggestions. We merged all the citations throughout the manuscript according to the journal style (e.g. line 44: “[1-3]”; line 53: “[9,10]”)
  2. Line 78 change as  mir-133, and miR-1; MALAT1, and Airn
  • We thank for the suggestions. We changed as suggested

Line 90: “lncRNAs (e.g., H19, NEAT1, MALAT1, and AIRN).”

  1. Line 124 L is normal as L
  • We changed L as L

Line 140: “to evaluate the combined effect of Hippophae rhamnoides L. (sea buckthorn, SBU)”

  1. Line 127 change PTGS2, and CCL2
  • We changed as suggested

Line143: “identified key genes (SERPINE1, EGFR, PTGS2, and CCL2) and…”

  1. Line 247 delete (2021)

-We thank the reviewer for spotting this misplaced text. We realized that the paragraph containing this error was indeed part of a previous draft, which had been wrongly included in the final version. This line does not exist anymore. The paragraph has been corrected with the final edited version.

References

  1. In all references the numbers are duplicated please delete the duplicate
  • We thank the reviewer for noticing this error. We believe this was due to the formatting into the journal style. We have deleted all the duplicates
  1. Reference 22, 43 use abbreviations of journal
  • We thank for the comment. We have corrected the reference list, adding the missing information.

Reviewer 4 Report

Comments and Suggestions for Authors

This is a timely, well-structured, and comprehensive review that addresses a highly relevant topic at the intersection of cardiometabolic disease, fibrosis, and epigenetic regulation. The authors successfully synthesise a large body of recent evidence to build a compelling narrative: that the cardioprotective effects of various metabolic modulators are mediated, in significant part, through their ability to reprogram the non-coding RNA (ncRNA) landscape. The paper is suitable for a broad audience in metabolic and cardiovascular research and has significant value for both basic scientists and clinically oriented readers. The following comments must be addressed by the authors.

Major Comments and Suggestions for Revision

  1. Fulfil the Promise on circRNAs: The abstract introduces circular RNAs (circRNAs) as a key focus, but they are not discussed in the review. Action: Either add a dedicated section on circRNAs or explicitly state that their investigation is an emerging field and a current knowledge gap.
  2. Include a Unifying Visual Summary: The complex mechanisms would be significantly clarified by a graphical abstract or schematic figure. Action: Create a figure illustrating how the different drug classes converge on ncRNA nodes to inhibit common fibrotic pathways (e.g., TGF-β).
  3. Address Apparent Contradictions: Resolve conflicting findings, such as MALAT1 being described as both pro-fibrotic and upregulated by a potentially beneficial drug (Pioglitazone). Action: Provide brief commentary on the context-dependent nature of ncRNA biology to reconcile these points.
  4. Explore Exosomal Mechanisms: The introduction mentions extracellular vesicles but doesn't explore them as a potential mechanism of drug action. Action: Integrate a discussion on whether these therapies might alter exosomal ncRNA cargo, a novel and highly relevant potential mechanism.
  5. Balance the Depth of Sections: The depth of analysis is inconsistent across drug classes. Action: Strengthen the PPARγ and GLP-1 RA sections to match the mechanistic detail provided for Metformin and SGLT2 inhibitors.
  6. Refine the Biomarker Discussion: The potential for ncRNAs as biomarkers is mentioned, but underdeveloped. Action: Expand on this in the "Therapeutic Implications" section with specific examples of circulating ncRNAs (e.g., from the clinical studies mentioned) and their practical utility.
  7. Correct Minor Errors: There is a likely typo in Table 1 where the lncRNA "Aim" should probably be corrected to "AIM" (Antioxidant Inducing Mitochondrial DNA).
  8. Strengthen the Clinical Translation Narrative: While clinical observations are noted, the path to application could be clearer. Action: Briefly discuss the challenges and next steps (e.g., standardisation of assays, cohort validation) in moving these ncRNA biomarkers and mechanisms toward clinical use.
  9. Enhance the Future Directions Scope: The conclusion can be more forward-looking. Action: Suggest specific future research avenues, such as exploring drug combinations based on ncRNA profiles or developing ncRNA-based therapeutics mimetic of these drug effects.
  10. Ensure Consistent Terminology: Use consistent formatting for non-coding RNA abbreviations (e.g., lncRNA instead of "IncRNA") throughout the manuscript to maintain professionalism. And also abbreviation at first instance is essential.
  11. A few minor grammatical syntax errors must be fixed throughout the manuscript.

Author Response

  1. Fulfil the Promise on circRNAs: The abstract introduces circular RNAs (circRNAs) as a key focus, but they are not discussed in the review. Action: Either add a dedicated section on circRNAs or explicitly state that their investigation is an emerging field and a current knowledge gap.

  • We thank the reviewer for this constructive suggestion. We added Section 2.6 discussing current evidence on cardiometabolic therapies and circRNAs, and highlighting the knowledge gap regarding their role in cardiac fibrosis.

2. Include a Unifying Visual Summary: The complex mechanisms would be significantly clarified by a graphical abstract or schematic figure. Action: Create a figure illustrating how the different drug classes converge on ncRNA nodes to inhibit common fibrotic pathways (e.g., TGF-β).

- We appreciate this helpful recommendation. We created Figure 1, summarizing how metabolic drugs modulate ncRNAs and converge on molecular pathways regulating cardiovascular

  1. Address Apparent Contradictions: Resolve conflicting findings, such as MALAT1 being described as both pro-fibrotic and upregulated by a potentially beneficial drug (Pioglitazone). Action: Provide brief commentary on the context-dependent nature of ncRNA biology to reconcile these points.
  • We thank the reviewer for raising this important point. We reviewed the literature and noted MALAT1’s role in inflammation, suggesting its relevance to cardiac fibrosis in metabolic disease. We also included evidence on lncR-30245, which is inhibited by Rosiglitazone and reduces fibrosis (highlighted in the text from line 261).

  1. Explore Exosomal Mechanisms: The introduction mentions extracellular vesicles but doesn't explore them as a potential mechanism of drug action. Action: Integrate a discussion on whether these therapies might alter exosomal ncRNA cargo, a novel and highly relevant potential mechanism.
  • We are grateful for this insightful comment. We expanded Section 3 ('Therapeutic Implications') to discuss how cardiometabolic therapies may influence exosomal ncRNA delivery and its potential for therapeutic applications.

  1. Balance the Depth of Sections: The depth of analysis is inconsistent across drug classes. Action: Strengthen the PPARγ and GLP-1 RA sections to match the mechanistic detail provided for Metformin and SGLT2 inhibitors.
  • We thank the reviewer for this observation. The variation in detail across sections reflects the uneven availability of published data. Metformin and SGLT2 inhibitors are well studied, while fewer data exist for PPARγ agonists, GLP-1 receptor agonists, and fatty acid oxidation inhibitors.

  1. Refine the Biomarker Discussion: The potential for ncRNAs as biomarkers is mentioned, but underdeveloped. Action: Expand on this in the 'Therapeutic Implications' section with specific examples of circulating ncRNAs (e.g., from the clinical studies mentioned) and their practical utility.
  • We appreciate the reviewer’s valuable suggestion. We expanded Section 3 to include recent evidence on circulating and exosomal ncRNAs as biomarkers of therapy response and patient stratification).

  1. Correct Minor Errors: There is a likely typo in Table 1 where the lncRNA 'Aim' should probably be corrected to 'AIM' (Antioxidant Inducing Mitochondrial DNA).
  • We thank the reviewer for noting this error. We corrected 'Airn' to 'AIRN' in both Table 1 and the main text.

  1. Strengthen the Clinical Translation Narrative: While clinical observations are noted, the path to application could be clearer. Action: Briefly discuss the challenges and next steps (e.g., standardisation of assays, cohort validation) in moving these ncRNA biomarkers and mechanisms toward clinical use.
    Enhance the Future Directions Scope: The conclusion can be more forward-looking. Action: Suggest specific future research avenues, such as exploring drug combinations based on ncRNA profiles or developing ncRNA-based therapeutics mimetic of these drug effects.
  • We are grateful for this constructive recommendation. We expanded the Conclusions section (line 454 onwards, as highlighted in the text)  to address translational challenges and outline future research directions.

  1. Ensure Consistent Terminology: Use consistent formatting for non-coding RNA abbreviations (e.g., lncRNA instead of 'IncRNA') throughout the manuscript to maintain professionalism. And also abbreviation at first instance is essential.
  • We thank the reviewer for highlighting this need. We harmonized terminology and ensured all abbreviations are defined at first mention.

  1. A few minor grammatical syntax errors must be fixed throughout the manuscript.
  • We appreciate the reviewer’s careful reading. We thoroughly revised the manuscript for grammar and syntax.

Reviewer 5 Report

Comments and Suggestions for Authors

Manuscript ID: metabolites-3843783

Review report: Rejected (The reviw lacks critical analysis of the existing data and information)

This review paper highlights how metabolic modulators such as metformin, SGLT2 inhibitors, PPARγ agonists, GLP-1 receptor agonists and fatty acid oxidation inhibitors can influence non-coding RNA expression, thereby reducing cardiac fibrosis and improving myocardial function. The manuscript may be further improved by following suggestions:

  1. The word already mentioned in the title should not be repeated in keyword, select appropriate keyword relevant to the study.
  2. Provide details global statistics with recent references on cardiovascular fibrosis, current treatment option, survivability/mortality rate, & research gap.
  3. The review is comprehensive but could benefit from a more systematic methodology section outlining inclusion/exclusion criteria for literature selection.
  4. The narrative sometimes reads as descriptive rather than critical; adding comparative analysis of drug classes would strengthen the manuscript.
  5. Figures or schematic diagrams summarizing drug-ncRNA-fibrosis pathways could improve clarity.
  6. Some sections are disproportionately detailed (e.g., metformin) while others (e.g., PPARγ agonists, GLP-1 agonists) feel underdeveloped.
  7. The review could address potential limitations, such as variability in ncRNA measurement techniques across studies.
  8. Clinical translation is discussed, but the gap between preclinical and clinical findings should be emphasized more.
  9. Discussion of circRNAs is limited compared to miRNAs and lncRNAs, despite their growing importance.
  10. References are abundant, but some statements would benefit from more recent or higher-impact clinical trial citations.
  11. The role of extracellular vesicle-mediated ncRNA transport is briefly mentioned but deserves a fuller discussion given its therapeutic potential.
  12. The conclusion could be more forward-looking by outlining specific future research priorities and challenges in integrating ncRNA-based biomarkers into routine cardiometabolic care.
  13. While the paper provides a broad overview of existing literature, it tends to summarize studies rather than critically evaluate them. A good review should not only compile findings but also identify gaps, inconsistencies & methodological limitations in previous work. Currently, the paper does not sufficiently assess the strength of evidence, compare contrasting viewpoints or highlight areas where research is lacking or inconclusive. Strengthening the critical analysis would improve the depth and scholarly value of the review, making it more useful for readers seeking insight rather than just description.
Comments on the Quality of English Language

May be improved.

Author Response

  1. The word already mentioned in the title should not be repeated in keyword, select appropriate keyword relevant to the study.
  • We thank the reviewer for this helpful remark. Keywords were updated to: fibrotic remodeling; metabolic modulators; microRNAs; long non-coding RNAs; epigenetic regulation; cardiometabolic syndromes; therapeutic targets.

  1. Provide details global statistics with recent references on cardiovascular fibrosis, current treatment option, survivability/mortality rate, & research gap.
  • We are grateful for this important suggestion. We added a concise overview of cardiac fibrosis in the Introduction with recent references (please see highlighted text within the Introduction from line 59). As fibrosis usually develops secondary to other cardiovascular diseases, survival and mortality data cannot be attributed to fibrosis alone—this limitation is now emphasized.

  1. The review is comprehensive but could benefit from a more systematic methodology section outlining inclusion/exclusion criteria for literature selection.
  • We thank the reviewer for the comment and clarify accordingly. As this is not a systematic review, inclusion/exclusion criteria were not applicable.

  1. The narrative sometimes reads as descriptive rather than critical; adding comparative analysis of drug classes would strengthen the manuscript.
  • We appreciate the reviewer’s constructive input. We added a comparative analysis in the Conclusion, highlighting similarities and differences across drug classes (please see highlighted tex from line 471 to 481).

  1. Figures or schematic diagrams summarizing drug-ncRNA-fibrosis pathways could improve clarity.
  • We thank the reviewer for this excellent suggestion. A schematic summary has been added as Figure 1, illustrating how metabolic drugs regulate ncRNAs and fibrotic pathways.

  1. Some sections are disproportionately detailed (e.g., metformin) while others (e.g., PPARγ agonists, GLP-1 agonists) feel underdeveloped.
  • We are grateful for this observation. Section depth reflects the uneven volume of available evidence: Metformin and SGLT2 inhibitors are extensively studied, while PPARγ agonists, GLP-1 agonists, and FAO inhibitors are less explored in fibrosis.

  1. The review could address potential limitations, such as variability in ncRNA measurement techniques across studies.
    Clinical translation is discussed, but the gap between preclinical and clinical findings should be emphasized more.
  • We thank the reviewer for pointing this out. The Conclusion was expanded to discuss methodological variability, translational challenges, and research needs.

  1. Discussion of circRNAs is limited compared to miRNAs and lncRNAs, despite their growing importance.
  • We appreciate the reviewer’s helpful recommendation. A new Section entitled

    "2.6. Cardiometabolic therapies and circularRNAs: a current knowledge gap (line 330 to 353)"

    was added to cover circRNAs and their potential role in fibrosis modulation by cardiometabolic therapies.

  1. References are abundant, but some statements would benefit from more recent or higher-impact clinical trial citations.
  • We thank the reviewer for highlighting this point. References were updated with recent and high-impact studies; the majority are from the last five years.

  1. The role of extracellular vesicle-mediated ncRNA transport is briefly mentioned but deserves a fuller discussion given its therapeutic potential.
  • We are grateful for this valuable suggestion. Section 3 (Therapeutic Implications) was expanded (please see highlighted text line 399 -443) to cover exosomal ncRNA delivery and therapeutic potential.

  1. The conclusion could be more forward-looking by outlining specific future research priorities and challenges in integrating ncRNA-based biomarkers into routine cardiometabolic care.
    While the paper provides a broad overview of existing literature, it tends to summarize studies rather than critically evaluate them. A good review should not only compile findings but also identify gaps, inconsistencies & methodological limitations in previous work.
  • We thank the reviewer for this important recommendation. The Conclusions were expanded to include a more critical appraisal of the literature, highlight gaps, and propose future research priorities.

Reviewer 6 Report

Comments and Suggestions for Authors

The review manuscript titled :Cardiometabolic Therapies Shape Non-Coding RNA Landscapes in Cardiovascular summarized an important and timely topic with strong scientific value. The manuscript is well designed and provides a clear focus on the involvement of different metabolic pathway molecules in cardiovascular fibrosis. The references cited are up to date. However, several issues should be addressed before publication:

  1. Methodological Details

    • The methodology is only briefly mentioned in the abstract and is not adequately described in the main text. The authors should clearly state the number of studies reviewed, the databases or resources used, the search criteria applied, and how relevant literature was selected.

  2. Abstract Consistency

    • The keyword epigenetic regulation is listed but not mentioned in the abstract, despite being referenced 12 times throughout the manuscript. This concept should be incorporated into the abstract for consistency and completeness.

  3. Abbreviations

    • Several abbreviations (e.g., SGLT2, PPARγ, GLP-1, MALAT1, TGF-β) appear without being defined at first use. These should be spelled out in full upon initial appearance to ensure clarity for readers.

  4. Reference Issues

    • Duplicate references are present in lines 217 and 219. These need correction.

  5. Punctuation

    • Line 227 is missing a full stop and lines 246 and 320 without correct comma or stop.

Author Response

  1. The methodology is only briefly mentioned in the abstract and is not adequately described in the main text. The authors should clearly state the number of studies reviewed, the databases or resources used, the search criteria applied, and how relevant literature was selected.
  • We thank the reviewer for the comment and clarify accordingly throughout the text including the Abstract and Introduction sections. As this is a narrative review rather than a systematic review, methodology details such as database search criteria are not applicable.

  1. The keyword epigenetic regulation is listed but not mentioned in the abstract, despite being referenced 12 times throughout the manuscript. This concept should be incorporated into the abstract for consistency and completeness.
  • We appreciate this helpful remark. The abstract was updated to include the term 'epigenetic regulators'.

  1. Several abbreviations (e.g., SGLT2, PPARγ, GLP-1, MALAT1, TGF-β) appear without being defined at first use. These should be spelled out in full upon initial appearance to ensure clarity for readers.
  • We thank the reviewer for drawing attention to this point. Abbreviations were standardized and defined at first mention throughout the manuscript.

  1. Duplicate references are present in lines 217 and 219. These need correction.
  • We appreciate the reviewer for noticing this error. Duplicate references were removed.

  1. Line 227 is missing a full stop and lines 246 and 320 without correct comma or stop.
  • We thank the reviewer for pointing out these mistakes. Punctuation errors were corrected.

Round 2

Reviewer 1 Report

Comments and Suggestions for Authors

The authors have answered most of my queries. But it would be better if they address the below mentioned comments too.

The current tables mention findings from cell, animal and human studies within the same rows which is a bit confusing. In my view, its better to organize the tables into separate preclinical and clinical sections.

Please strengthen discussion on technical variability and limitations of ncRNA biomarkers. I understand it’s a narrative review. But this would strengthen the MS.

A minor further refinement of english is necessary

Author Response

  1. The authors have answered most of my queries. But it would be better if they address the below mentioned comments too.The current tables mention findings from cell, animal and human studies within the same rows which is a bit confusing. In my view, its better to organize the tables into separate preclinical and clinical sections.

  • We thank the reviewer for this suggestion. We believe that the information included in the Table, as recommended by Reviewer 2 during the first round of revision, is indeed valuable for the readers. It clearly specifies the experimental context from which the presented evidence originates. We agreed with this comment, and in our view, the information provided is complete, informative, and well-aligned with the scope of the proposed review.

  1. Please strengthen discussion on technical variability and limitations of ncRNA biomarkers. I understand it’s a narrative review. But this would strengthen the MS.
  • We thank for the helpful comment. We provided an additional paragraph in the section 3 (lines 416-444) explaining the current limitations for using ncRNA measure as reliable biomarkers. We added relevant references on the matter, focusing on miRNAs which are to date the more extensively tested ncRNAs to be used as biomarkers for disease progression as well as therapeutic effects of molecules in general. We hope the text provided will indeed strengthen the manuscript.

  1. A minor further refinement of english is necessary
  • We thank the reviewer for thorough assessment of English language usage. The manuscript has been thoroughly checked for grammatical accuracy, clarity, and stylistic consistency. In particular, the usage of tenses (present perfect instead of simple past as appropriate) has been assessed and made consistent through the paragraphs. Please find the grammatical corrections highlighted in yellow with red text, in order to distinguish them from the amendments to the text made in the first round of revision.

Reviewer 5 Report

Comments and Suggestions for Authors

Author has done significant improvement, it can be accepted.

Comments on the Quality of English Language

May be improved.

Author Response

1. Author has done significant improvement, it can be accepted.

- We thank the reviewer for acknowledging our effort to improve the quality of our manuscript. 

Comments on the Quality of English Language

1. May be improved.

- The manuscript has been thoroughly checked for grammatical accuracy, clarity, and stylistic consistency. In particular, the usage of tenses (present perfect instead of simple past as appropriate) has been assessed and made consistent through the paragraphs.